# HYPERFIELDS: TOWARDS ZERO-SHOT GENERATION OF NERFS FROM TEXT

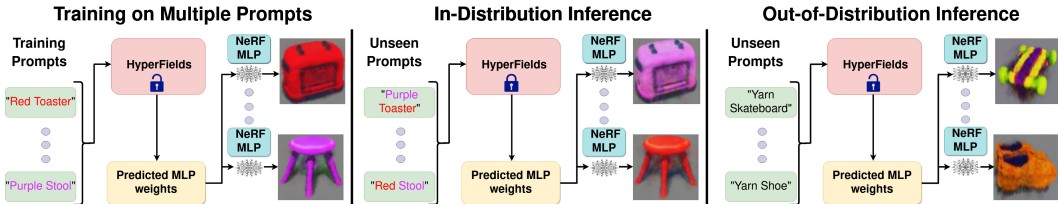

Figure 1: HyperFields is a hypernetwork that learns to map text to the space of weights of Neural Radiance Fields (first column). On learning such a mapping, HyperFields is capable of generating in-distribution scenes (unseen during training) in a feed forward manner (second column), and for unseen out-of-distribution prompts HyperFields can be fine-tuned to yield scenes respecting prompt semantics with just a few gradient steps (third column).

## ABSTRACT

We introduce HyperFields, a method for generating text-conditioned Neural Radiance Fields (NeRFs) with a single forward pass and (optionally) some fine-tuning. Key to our approach are: (i) a dynamic hypernetwork, which learns a smooth mapping from text token embeddings to the space of NeRFs; (ii) NeRF distillation training, which distills scenes encoded in individual NeRFs into one dynamic hypernetwork. These techniques enable a single network to fit over a hundred unique scenes. We further demonstrate that HyperFields learns a more general map between text and NeRFs, and consequently is capable of predicting novel in-distribution and out-of-distribution scenes — either zero-shot or with a few finetuning steps. Finetuning HyperFields benefits from accelerated convergence thanks to the learned general map, and is capable of synthesizing novel scenes 5 to 10 times faster than existing neural optimization-based methods. Our ablation experiments show that both the dynamic architecture and NeRF distillation are critical to the expressivity of HyperFields.

## 1 INTRODUCTION

Recent advancements in text-to-image synthesis methods, highlighted by the works of Ramesh et al. (2021); Yu et al. (2022), have ignited interest in achieving comparable success in the field of text-to-3D synthesis. This interest has grown in tandem with the emergence of Neural Radiance Fields (NeRFs) (Mildenhall et al., 2020; Yu et al., 2021b; Jain et al., 2021), which is a popular 3D representation for this task, due to their ability to robustly depict complex 3D scenes.

To date, most text-conditioned 3D synthesis methods rely on either text-image latent similarity matching or diffusion denoising, both which involve computationally intensive per-prompt NeRF optimization (Jain et al., 2022; Poole et al., 2022; Lin et al., 2022). Extending these methods to bypass the need for per-prompt optimization remains a non-trivial challenge.

We propose to solve this problem through a hypernetwork-based neural pipeline, in which a single hypernetwork (Ha et al., 2016b) is trained to generate the weights of individual NeRF networks, each corresponding to an unique scene. Once trained, this hypernetwork is capable of efficiently producing the weights of NeRFs corresponding to novel prompts, either through a single forward pass or with minimal fine-tuning. Sharing the hypernetwork across multiple

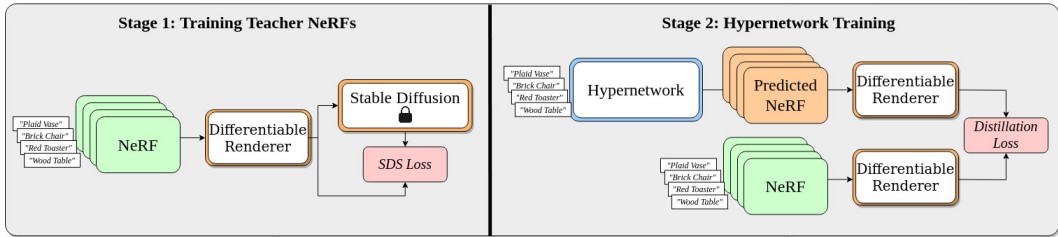

Figure 2: **Overview.** Our training pipeline proceeds in two stages. **Stage 1:** We train a set of single prompt text-conditioned teacher NeRFs using Score Distillation Sampling. **Stage 2:** We distill these single scene teacher NeRFs into the hypernetwork, through a photometric loss between the renders of the hypernetwork with the teacher network, which we dub our *distillation loss.*

training scenes enables effective transfer of knowledge to new scenes, leading to better generalization and faster convergence. However, we find that a naive hypernetwork design is hard to train.

Our method, *HyperFields*, overcomes these challenges through several design choices. We propose predicting the weights of each layer of the NeRF network in a *progressive* and *dynamic* manner. Specifically, we observe that the intermediate (network) activations from the hypernetwork-predicted NeRF can be leveraged to guide the prediction of subsequent NeRF weights effectively.

To enhance the training of our hypernetwork, we introduce an alternative distillation-based framework rather than the Score Distillation Sampling (SDS) used in Poole et al. (2022); Wang et al. (2022). We introduce NeRF distillation, in which we first train individual text-conditioned NeRF scenes (using SDS loss) that are used as teacher NeRFs to provide fine-grained supervision to our hypernetwork (see Fig. 2). The teacher NeRFs provide exact colour and geometry labels, eliminating any potentially noisy training signals.

Our NeRF distillation framework allows for training HyperFields on a much larger set of scenes than with SDS, scaling up to 100 different scenes without any degradation in scene quality. A potential explanation for this is that SDS loss exhibits high variance in loss signals throughout different sampling steps. This instability in the loss likely contributes to the challenge of training the hypernetwork on multiple scenes.

Once trained, our model can synthesize novel in-distribution NeRF scenes in a single forward pass (Fig. 1, second column) and enables accelerated convergence for out-of-distribution scenes, requiring only a few fine-tuning steps (Fig. 1, third column). We clarify our use of the terms "in-distribution" and "out-of-distribution" in Sections 4.1 and 4.2 respectively. These results suggest that our method learns a semantically meaningful mapping. We justify our design choices through ablation experiments which show that both dynamic hypernetwork conditioning and NeRF distillation are critical to our model's expressivity.

Our successful application of dynamic hypernetworks to this difficult problem of generalized text-conditioned NeRF synthesis suggests a promising direction for future work on generalizing and parameterizing neural implicit functions through other neural networks.

## 2 BACKGROUND AND RELATED WORK

Our work combines several prominent lines of work: neural radiance fields, score-based 3D synthesis, and learning function spaces using hypernetworks.

### 2.1 3D REPRESENTATION VIA NEURAL RADIANCE FIELDS

There are many competing methods of representing 3D data in 3D generative modeling, such as point-clouds (Nichol et al., 2022; Zhou et al., 2021), meshes (Michel et al., 2021; Hong et al., 2022; Metzer et al., 2022; Zeng et al., 2022), voxels (Sanghi et al., 2021; 2022), and signed-distance fields (Wang et al., 2021; Yariv et al., 2021; Esposito et al., 2022). This work explores the popular representation of 3D scenes by Neural Radiance Fields (NeRF) (Mildenhall et al., 2020; Xie et al., 2021; Gao et al., 2022). NeRFs were originally introduced to handle the task of multi-view

reconstruction, but have since been applied in a plethora of 3D-based tasks, such as photo-editing, 3D surface extraction, and large/city-scale 3D representation (Gao et al., 2022).

There have been many improvements on the original NeRF paper, especially concerning training speed and fidelity (Chen et al., 2022a;b; Müller et al., 2022; Sun et al., 2021; Yu et al., 2021a). HyperFields uses the multi-resolution hash grid introduced in InstantNGP (Müller et al., 2022).

## 2.2 Score-Based 3D Generation

While many works attempt to directly learn the distribution of 3D models via 3D data, others opt to use guidance from 2D images due to the significant difference in data availability. Such approaches replace the photometric loss in NeRF's original view synthesis task with a guidance loss. The most common forms of guidance in the literature are from CLIP (Radford et al., 2021) or a frozen, text-conditioned 2D diffusion model. The former methods seek to minimize the cosine distance between the image embeddings of the NeRF's renderings and the text embedding of the user-provided text prompt (Jain et al., 2022; Chen et al., 2022a; Jain et al., 2021).

Noteworthy 2D diffusion-guided models include DreamFusion (Poole et al., 2022) and Score Jacobian Chaining (SJC) (Wang et al., 2022), which feed noised versions of images rendered from a predicted NeRF into a frozen text-to-image diffusion model (Imagen (Saharia et al., 2022) and StableDiffusion Rombach et al. (2021), respectively) to obtain what can be understood as a scaled Stein Score (Liu et al., 2016). Our work falls into this camp, as we rely on score-based gradients derived from StableDiffusion to train the NeRF models which guide our hypernetwork training.

Specifically, we use the following gradient motivated in DreamFusion:

$$\nabla_\theta \mathcal{L}(\phi, g(\theta)) \triangleq \mathbb{E}_{t,c} \left[ w(t)(\hat{\epsilon}_\phi(z_t; y, t) - \epsilon)\frac{\partial x}{\partial \theta} \right] \tag{1}$$

which is similar to the gradient introduced in SJC, the key difference being SJC directly predicts the noise score whereas DreamFusion predicts its residuals. We refer to optimization using this gradient as *Score Distillation Sampling* (SDS), following the DreamFusion authors. More recently following works are directed at improving 3D generation quality (Wang et al., 2023; Metzer et al., 2023; Chen et al., 2023), while our focus is on an orthogonal problem of improving generalization and convergence of text to 3D models.

**Connections to ATT3D:** We note that our work is concurrent and independent of ATT3D (Lorraine et al., 2023). We are similar in that we both train a hypernetwork to generate NeRF weights for a set of scenes during training and generalize to novel in-distribution scenes without any test time optimization. However also demonstrate accelerated convergence to novel out-of-distribution scenes, requiring only a few optimization steps. We clarify on our use of terms in-distribution and out-of-distribution in Sec. 4.1 and 4.2 respectively.

We primarily differ in the input and the application of the hypernetwork. Our hypernetwork generates the MLP weights of the NeRF, while ATT3D outputs the weights of the hash grid in their InstantNGP model. We condition our hypernetwork on the text prompt and activations of the generated NeRF MLP (Sec. 3), while ATT3D is conditioned on just the text prompt.

Finally, ATT3D is built on Magic3D (Lin et al., 2022) which is a proprietary and more powerful text-to-3D model than the publicly available stable DreamFusion model (Tang, 2022) that we use. In contrast we plan to open-source our code and trained model.

## 2.3 Hypernetworks

Hypernetworks are networks that are used to generate weights of other networks which perform the actual task (task performing network) (Ha et al., 2016a). Many works attempt to use hypernetworks as a means to improve upon conditioning techniques. Among these, some works have explored applying hypernetworks to implicit 2d representations (Sitzmann et al., 2020; Perez et al., 2017; Alaluf et al., 2021), and 3D representations (Sitzmann et al., 2019; 2021; Chiang et al., 2021). Very few works apply hypernetworks to radiance field generation. Two notable ones are HyperDiffusion and Shape-E, which both rely on denoising diffusion for generation (Erkoç et al., 2023; Jun & Nichol, 2023). HyperDiffusion trains an unconditional generative model which diffuses over

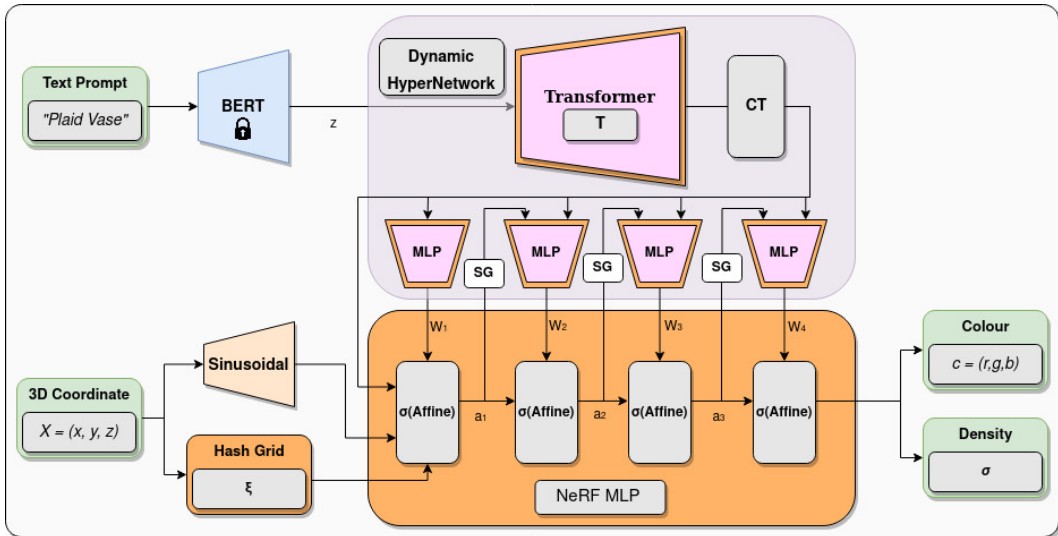

Figure 3: The input to the HyperFields system is a text prompt, which is encoded by a pre-trained text encoder (frozen BERT model). The text latents are passed to a Transformer module, which outputs a conditioning token (CT). This conditioning token (which supplies scene information) is used to condition each of the MLP modules in the hypernetwork. The first hypernetwork MLP (on the left) predicts the weights $W_1$ of the first layer of the NeRF MLP. The second hypernetwork MLP then takes as input both the CT and $a_1$, which are the *activations* from the first predicted NeRF MLP layer, and predicts the weights $W_2$ of the second layer of the NeRF MLP. The subsequent scene-conditioned hypernetwork MLPs follow the same pattern, taking the activations $a_{i-1}$ from the previous predicted NeRF MLP layer as input to generate weights $W_i$ for the $i^{th}$ layer of the NeRF MLP. We include stop gradients (SG) so stabilize training.

sampled NeRF weights, and thus cannot do text-conditioned generation. Shap-E diffuses over latent codes which are then mapped to weights of a NeRF MLP, and requires teacher point clouds to train. Due to the memory burden of textured point clouds, scene detail is not well represented in Shap-E. Both of these methods have the same limitations of slow inference due to denoising sampling. In contrast, our method predicts NeRF weights dynamically conditioned on the 1) text prompt, 2) the sampled 3D coordinates, and 3) the previous NeRF activations.

An interesting class of hypernetworks involve models conditioned on the activations or inputs of the task-performing network (Chen et al., 2020). These models take the following form: let $h, g$ be the hypernetwork and the task performing network respectively. Then $W = h(a)$, where $W$ acts as the weights of $g$ and $a$ is the activation from the previous layer of $g$ or the input to the first layer of $g$. These are called dynamic hypernetworks as the predicted weights change dynamically with respect to the layer-wise signals in $g$. In the static hypernetwork case, $W = h(e)$ where W is still the weights of $g$ but $e$ is learned and is independent of the input to $g$. Our work explores the application of dynamic hypernetworks to learning a general map between text and NeRFs.

## 3 METHOD

Our method consists of two key innovations, the dynamic hypernetwork architecture and NeRF distillation training. We discuss each of these two components in detail below.

### 3.1 DYNAMIC HYPERNETWORK

The dynamic hypernetwork consists of the Transformer $\mathcal{T}$ and MLP modules as given in figure 3. The sole input to the dynamic hypernetwork is the scene information represented as a text description. The text is then encoded by a frozen pretrained BERT model, and the text embedding $z$ is processed by $\mathcal{T}$. Let conditioning token CT = $\mathcal{T}(z)$ be the intermediate representation used to provide the current scene information to the MLP modules. Note that the text embeddings $z$ can come from any text encoder, though in our experiments we found frozen BERT embeddings to be the most performant.

In addition to conditioning token CT, each MLP module takes in the activations from the previous layer $a_{i-1}$ as input. Given these two inputs, the MLP module is tasked with generating parameters $W_i$ for the $i^{th}$ layer of the NeRF MLP. For simplicity let us assume that we sample only one 3D coordinate and viewing direction per minibatch, and let $h$ be the hidden dimension of the NeRF MLP. Then $a_{i-1} \in \mathbb{R}^{1 \times h}$. Now the weights $W_i \in \mathbb{R}^{h \times h}$ of the $i^{th}$ layer are given as follows:

$$W_i = \text{MLP}_i(CT, a_{i-1}) \tag{2}$$

The forward pass of the $i^{th}$ layer is:

$$a_i = W_i * a_{i-1} \tag{3}$$

where $a_i \in \mathbb{R}^{1 \times h}$ and * is matrix multiplication. This enables the hypernetwork MLPs to generate a different set of weights for the NeRF MLP that are best suited for each given input 3D point and viewing direction pair. This results in effectively a unique NeRF MLP for each 3D point and viewing direction pair.

However training with minibatch size 1 is impractical, so during training we sample a non-trivial minibatch size and generate weights that are best suited for the given minibatch as opposed to the above setting where we generate weights unique to each 3D coordinate and viewing direction pair.

In order to generate a unique set of weights for a given minibatch we do the following:

$$\overline{a}_{i-1} = \mu(a_{i-1}) \tag{4}$$
$$W_i = MLP_i(CT, \overline{a}_{i-1}) \tag{5}$$

Where $\mu(.)$ averages over the minibatch index. So if the minibatch size is $n$, then $a_{i-1} \in R^{n \times h}$, and $\overline{a}_{i-1} \in \mathbb{R}^{1 \times h}$ and the forward pass is still computed as given in equation 3. This adaptive nature of the predicted NeRF MLP weights leads to the increased flexibility of the model. As shown in our ablation experiments in Fig. 6a, it is an essential piece to our model's large scene capacity.

## 3.2 NeRF Distillation

As shown in figure 2, we first train individual DreamFusion NeRFs on a set of text prompts, following which we train the HyperFields architecture with supervision from these single-scene DreamFusion NeRFs.

The training routine is outlined in Algorithm 1, in which at each iteration, we sample $n$ prompts and a camera viewpoint for each of these text prompts (lines 2 to 4). Subsequently, for the $i^{th}$ prompt and camera viewpoint pair we render image $\mathcal{I}_i$ using the $i^{th}$ pre-trained teacher NeRF (line 5). We then condition the HyperFields network $\phi_{hf}$ with the $i^{th}$ prompt, and render the image $I_i^{'}$ from the $i^{th}$ camera view point (line 6). We use the image rendered by the pre-trained teacher NeRF as the ground truth supervision to HyperFields (line 7). For the same sampled $n$ prompts and camera viewpoint pairs, let $\mathcal{I}_1^{'}$ to $\mathcal{I}_n^{'}$ be the images rendered by HyperFields and $\mathcal{I}_1^{'}$ to $\mathcal{I}_n^{'}$ be the images rendered by the respective pre-trained teacher NeRFs. The distillation loss is given as follows:

$$\mathcal{L}_d = \frac{\sum_{i=1}^{n}(I_i - I_i^{'})^2}{n} \tag{6}$$

We observe through our ablations in Fig. 6b that this simple distillation scheme greatly helps HyperFields in learning to fit multiple text prompts simultaneously, as well as learn a more general mapping of text to NeRFs.

## 4 Results

We evaluate HyperFields by demonstrating its generalization capabilities, out-of-distribution convergence, amortization benefits, and ablation experiments. In Sec. 4.1 and Sec. 4.2 we evaluate the model's ability to synthesize novel scenes, both in and out-of-distribution. We quantify the amortization benefits of having this general model compared to optimizing individual NeRFs in Sec. 4.3. Finally, our ablations in Sec. 4.4 justify our design choices of dynamic conditioning and NeRF distillation training.

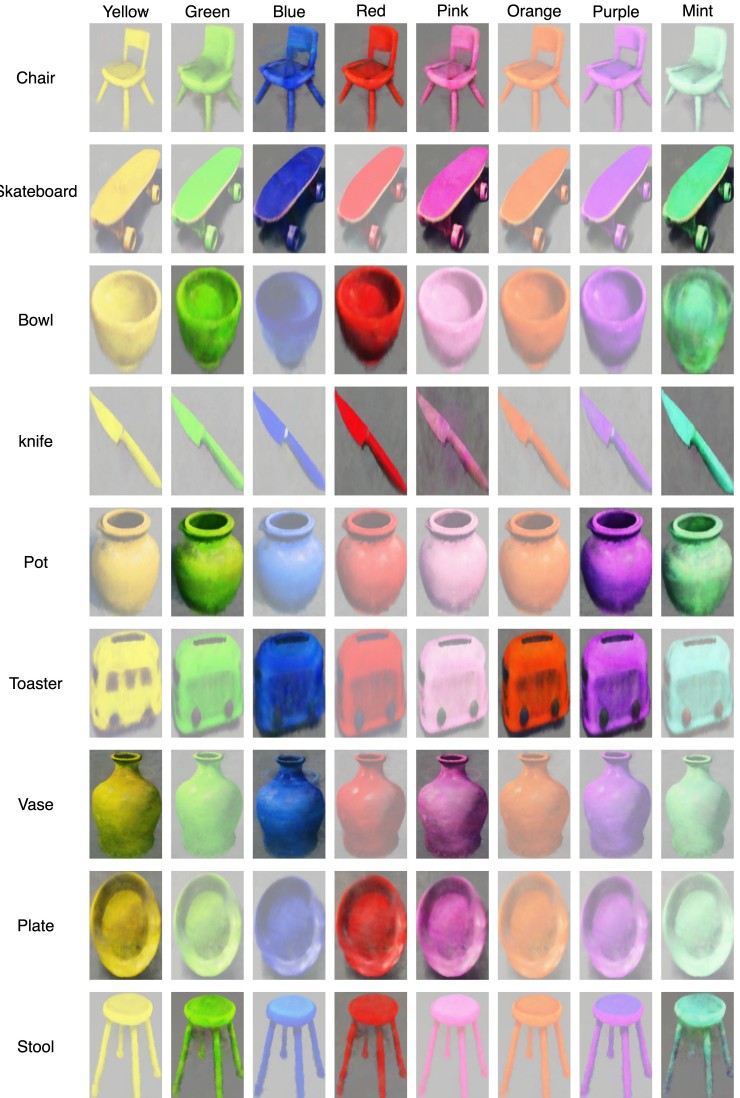

Figure 4: **Zero-Shot In-Distribution Generalization.**. During training, the model observes every individual shape and color, but we hold out a subset of color/shape combinations. During inference, the model generalizes by generating scenes for the held out combinations zero-shot. For example, "red chair" is an unseen combination, but the model is able to generalize from individual instances of "red" and "chair" from training. The faded scenes are generated from the training set, while the bright scenes are zero-shot predictions of the held-out prompts.

## 4.1 IN-DISTRIBUTION GENERALIZATION

Our method is able to train on a subset of the colour-shape combinations, and during inference predict the unseen colour-shape scenes *zero-shot, without any test time optimization*. Fig. 4 shows the results of training on a subset of combinations of 9 shapes and 8 colours, while holding out 3 colours for each shape. Our model generates NeRFs in a zero-shot manner for the held-out prompts (opaque scenes in Fig. 4) with quality nearly identical to the trained scenes.

We call this *in-distribution generalization* as both the shape and the color are seen during training but the inference scenes (opaque scenes in Fig.4) are novel because the combination of color and shape is unseen during training. Example: "Orange toaster" is a prompt the model has not seen during training, though it has seen the color "orange" and the shape "toaster" in its training set.

We quantitatively evaluate the quality of our zero-shot predictions with CLIP retrieval scores. The support set for the retrieval consists of all 72 scenes (27 unseen and 45 seen) shown in Fig. 4. In Table 1 we compute the top-$k$ retrieval scores by CLIP similarity. The table reports the average scores for Top-1, 3, 5, 6, and 10 retrieval, separated by unseen (zero-shot) and seen prompts. The similarity in scores between the unseen and seen prompts demonstrates that our model's zero-shot predictions are of similar quality to the training scenes with respect to CLIP similarity.

|  | Top-1 | Top-3 | Top-5 | Top-6 | Top-10 |
|---|---|---|---|---|---|
| Unseen | 57.1 | 85.7 | 85.7 | 90.4 | 95.2 |
| Seen | 69.5 | 88.1 | 94.9 | 94.9 | 96.6 |

Table 1: CLIP Retrieval Scores: We report the average retrieval scores for the scenes shown in Fig. 4. The small difference in scores between the seen and unseen scene prompts indicates that our zero-shot generations are of similar quality to the training scenes.

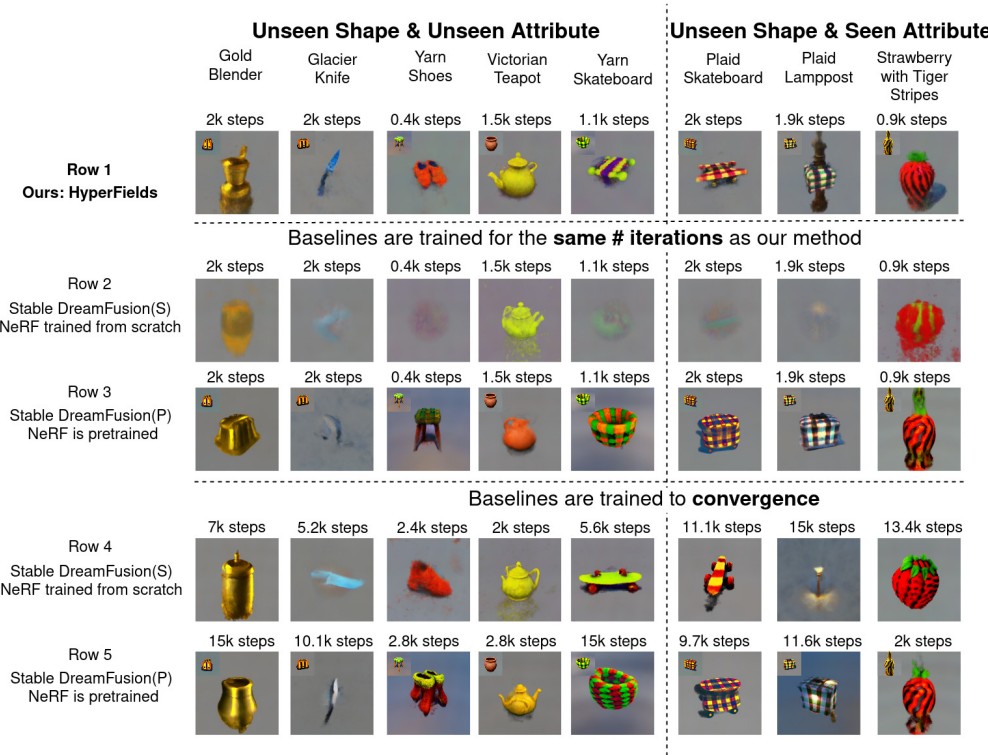

Figure 5: **Finetuning to out-of-distribution prompts: unseen shape and or unseen attribute.** Our method generates out-of-distribution scenes in at most 2k finetuning steps (row 1), whereas the baseline models are far from the desired scene at the same number of iterations (rows 2 and 3). When allowed to fine-tune for significantly longer (rows 4 and 5) the baseline generations are at best comparable to our model's generation quality, demonstrating that our model is able to adapt better to out-of-distribution scenes.

## 4.2 ACCELERATED OUT-OF-DISTRIBUTION CONVERGENCE

We further test HyperFields's ability to generate shapes and attributes that it has *not seen* during training. We call this *out-of-distribution inference* because the specified geometry and/or attribute are not within the model's training set.

We train our model on a rich source of prompts, across multiple semantic dimensions (material, appearance, shape). The list of prompts used is provided in the appendix material section D using NeRF distillation loss. Post training, we test our model on the prompts in Fig. 5. The prompts are grouped based on whether both shape and attribute are unseen (column 1, Fig. 5) or just the shape is unseen (column 2, Fig. 5). For example, in "gold blender" both material "gold" and shape "blender" are unseen during training.

Since these prompts contain geometry/attributes that are unseen during training, we do not expect high quality generation without any optimization. Instead, we demonstrate that fine-tuning the trained HyperFields model on SDS loss for the given the out-of-distribution prompt can lead to accelerated convergence especially when compared to the DreamFusion baselines.

We consider two baselines, 1) **Stable Dreamfusion (S):** Publicly available implementation of Dreamfusion trained from **S**cratch, 2) **Stable Dreamfusion (P):** Stable Dreamfusion model **P**re-trained on a semantically close scene and finetuned to the target scene. The motivation in using Stable Dreamfusion (P) is to have a pre-trained model as a point of comparison against HyperFields model.

We show out-of-distribution generation results for 8 different scenes in Fig. 5. The inset images in the upper left of row 1 of Fig. 5 are the scenes generated zero-shot by our method, *with no optimization*, when provided with the out-of-distribution prompt. The model chooses the *semantic nearest neighbour* from its training data as the initial guess for out-of-distribution prompts. For example, when asked for a "golden blender" and "glacier knife", our model generates a scene with "tiger striped toaster", which is the only related kitchenware appliance in the model sees during training. We pretrain the Stable Dreamfusion(P) baselines to the same scenes predicted by our model zero-shot. The pretrained scenes for Stable Dreamfusion(P) are given as insets in the upper left of row 3 and 5 in Fig. 5.

By finetuning on a small number of epochs for each out-of-distribution target scene using score distillation sampling, our method can converge much faster to the target scene than the baseline DreamFusion models. In row 2 and 3 of Fig. 5, we see that both Dreamfusion(S) and (P), barely learn the target shape for the same amount of training budget as our method. In rows 4 and 5 of Fig. 5 we let the baselines train to convergence, despite which the quality of the longer trained baseline scenes are worse or at best comparable to our model's generation quality. On average we see a 5x speedup in convergence. Additionally in Sec. E of the appendix we have a user study favourably comparing our generation to that of the baselines.

Importantly, DreamFusion(P) which is pre-trained to **the same zero-shot predictions of our model** is unable to be fine-tuned to the target scene as efficiently and at times get stuck in suboptimal local minima close to the initialization (see "yarn skateboard" row 3 and 5 in Fig. 5). This demonstrates that HyperFields learns a semantically meaningful mapping from text to NeRFs that cannot be arbitrarily achieved through neural optimization. We further explore the smoothness of this mapping through interpolation experiments in Sec. H of the appendix.

## 4.3 AMORTIZATION BENEFITS

The cost of pre-training HyperFields and individual teacher NeRFs is easily amortized in both in-distribution and out-of-distribution prompts. Training the teacher NeRFs is not an additional overhead; it's the cost of training a DreamFusion model on each of those prompts. The only overhead incurred by our method is the NeRF distillation training in stage 2 (Fig. 2), which takes roughly two hours. This overhead is offset by our ability to generate unseen combinations in a feedforward manner.

For comparison, the DreamFusion baseline takes approximately 30 minutes to generate each test scene in Fig. 4, totaling ∼14 hours for all 27 test scenes. In contrast, after the 2 hour distillation period, our model can generate all 27 test scenes in less than a minute, making it an order of magnitude faster than DreamFusion, even with the distillation overhead.

Our method's ability to converge faster to new out-of-distribution prompts leads to linear time-saving for each new prompt. This implies a practical use case of our model for rapid out-of-distribution scene generation in a real world setting. As shown in Fig. 5, the baseline's quality only begins to match ours after 3-5x the amount of training time.

## 4.4 ABLATIONS

The main contribution in our Dynamic Hypernetwork architecture is that the weights of the $i^{th}$ layer of the NeRF are generated as not only as a function of prompt but also as a function of activations from the $(i-1)^{th}$ layer. We show that using the activations from the previous layer in generating

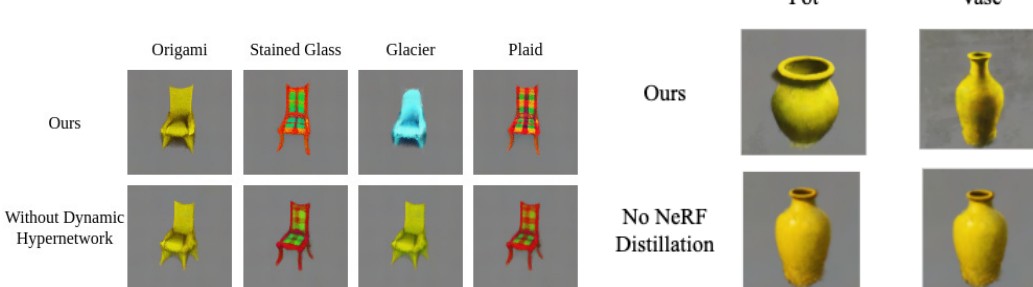

(a) **Dynamic Hypernet Packing.** Without dynamic conditioning ("Static Hypernet"), the hypernetwork packing ability is highly limited. We show 4 scenes packed using SDS, and the static hypernet collapses the origami/glacier attributes and stained glass/plaid attributes.

(b) **NeRF Distillation.** We compare our packing results when training the model from Fig. 4 with score distillation ("No NeRF Distillation") versus our NeRF distillation method ("Ours"). The iterative optimization of score distillation causes similar objects such as pot and vase to be guided towards the same common geometry.

subsequent weights is crucial. Without it our model's ability to pack multiple scenes is heavily reduced. In Fig. 6a row 2 ("Without Dynamic Hypernetwork"), shows that even in the simple case of 4 scenes the version of the hypernetwork *which does not use previous activations* for predicting the NeRF weights collapses the "glacier" and "origami" styles together and the "plaid" and "stained glass" styles together.

If we attempt to pack the dynamic hypernetwork using just Score Distillation Sampling (SDS) from DreamFusion, we experience a type of mode collapse in which the SDS optimization guides similar shapes towards the same common geometry. This also hinders the expressiveness of the hypernetwork and its ability to generate fine-grained, distinct geometry across different scenes. See Fig. 6b for an example of this mode collapse when attempting to train HyperFields with just SDS (no NeRF Distillation) over the set of scenes shown in Fig. 4.

### 4.5 LIMITATIONS

There are a few key limitations of HyperFields. First, the quality of the scenes used in training and consequently the quality of the generalization scenes is constrained by the quality of the teacher NeRFs. Any systematic artifacts in the teacher NeRFs will be learned by the hypernetwork. Second, the dynamic hypernetwork while allows for flexibility in representation of various scenes consumes a large amount of GPU RAM making our model trainable only on 48GB or hihger GPU RAMs. Finally, our model cannot generate scenes zero-shot without any optimization in an open vocabulary setting, and this is an interesting avenue for future research.

### 5 CONCLUSION

We present HyperFields, a novel framework for generalized text-to-NeRF synthesis, which can produce individual NeRF networks in a single feedforward pass. Our results highlight a promising step in learning a general representation of semantic scenes. Our novel dynamic hypernetwork architecture coupled with NeRF distillation learns an efficient mapping of text token inputs into a smooth and semantically meaningful NeRF latent space. Our experiments show that with this architecture we are able to fit over 100 different scenes in one model, and predict high quality unseen NeRFs either zero-shot or with a few finetuning steps. Comparing to existing work, our ability to train on multiple scenes greatly accelerates convergence of novel scenes. We plan on publishing our code and trained model shortly with an ArXiv release. In future work we would like to explore the possibility of generalizing the training and architecture to achieving zero-shot open vocabulary synthesis of NeRFs and other implicit 3D representations.

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

## A    MODEL DETAILS

**Baselines:** Our baseline is 6 layer MLP with skip connections every two layers. The hidden dimension is 64. We use an open-source re-implementation Tang (2022) of DreamFusion as both our baseline model and architecture predicted by HyperFields, because the original DreamFusion works relies on Google's Imagen model which is not open-source. Unlike the original DreamFusion, the re-implementation uses Stable Diffusion (instead of Imagen). We use Adam with a learning rate of 1e-4, with an epoch defined by 100 gradient descent steps.

**HyperFields:** The architecture is as described in Figure 2 in the main paper. The dynamic hypernetwork generates weights for a 6 layer MLP of hidden dimension 64. The transformer portion of the hypernetwork has 6 self-attention blocks, each with 12 heads with a head dimension of 16. We condition our model with BERT tokens, though we experiment with T5 and CLIP embeddings as well with similar but marginally worse success. Similar to the baseline we use Stable Diffusion for guidance, and optimize our model using Adam with the a learning rate of 1e-4. We will release open-source code of our project in a future revision of the paper.

We use the multiresolution hash grid developed in InstantNGP Müller et al. (2022) for its fast inference with low memory overhead, and sinusoidal encodings $\gamma$ to combat the known spectral bias of neural networks (Rahaman et al., 2018). The NeRF MLP has 6 layers (with weights predicted by the dynamic hypernetwork), with skip connections every two layers. The dynamic hypernetwork MLP modules are two-layer MLPs with ReLU non-linearities and the Transformer module has 6 self-attention layers. Furthermore, we perform adaptive instance normalization before passing the activations into the MLP modules of the dynamic hypernetwork and also put a stop gradient operator on the activations being passed into the MLP modules (as in figure 3).

## B    PACKING

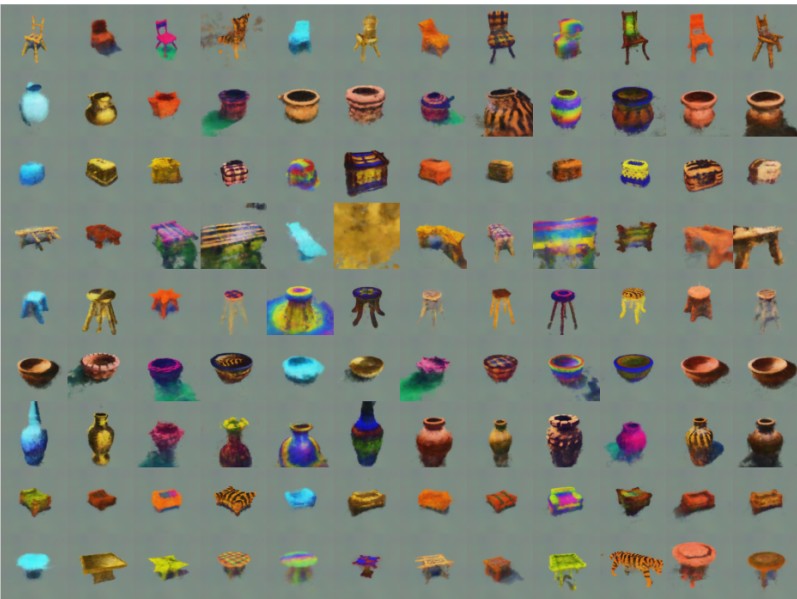

Figure 7: **Prompt Packing.** Our dynamic hypernetwork is able to pack 9 different objects across 12 different prompts for a total of 108 scenes. Dynamic hypetnetwork coupled with NeRF distillation enables packing these scenes into one network.

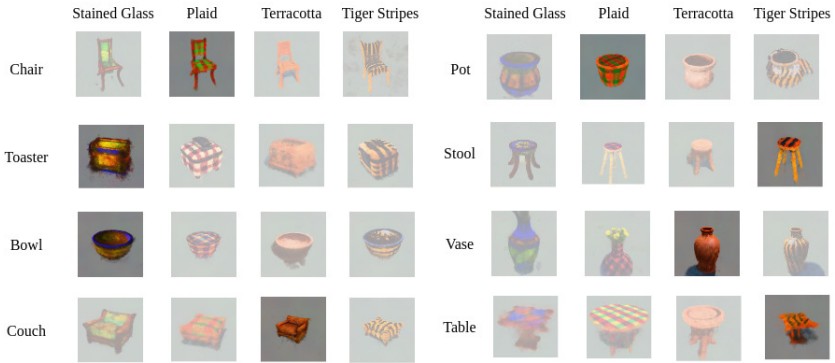

Figure 8: **Fine-Tuning In-Distribution: seen shape, seen attribute, unseen combination.** During training, the model observes every shape and color, but some combinations of shape and attribute remain unseen. During inference, the model generalizes by generating scenes that match prompts with previously unseen combinations of shape and attribute, with small amount of finetuning (atmost 1k steps).

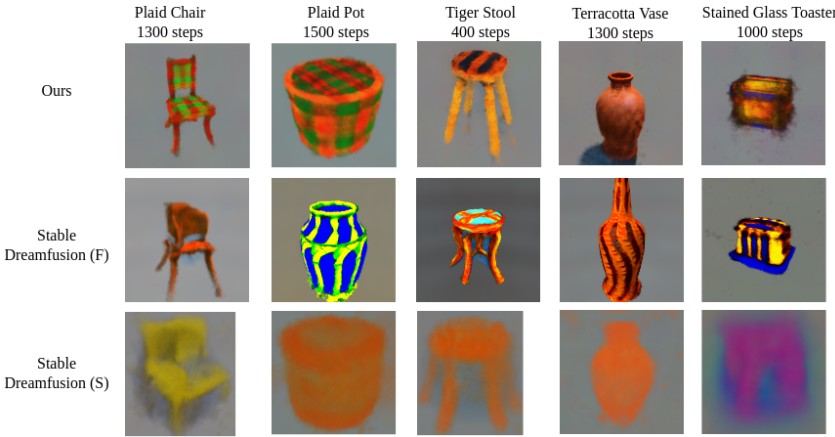

Figure 9: **Generalization Comparison.** We train a single HyperFields model and compare Stable DreamFusion. "Stable DreamFusion (F)" indicates finetuning from an initialized DreamFusion model. "Stable DreamFusion (S)" indicates the DreamFusion model trained from scratch. Zero-shot results and initializations are shown in the upper left of "Ours" and "Stable DreamFusion (F)", respectively. Above each column indicates the number of training epochs for each method add figures in the upper left.

## C  IN-DISTRIBUTION GENERALIZATION WITH COMPLEX PROMPTS

For additional attributes ("plaid", "Terracotta" etc.), our model produces reasonable zero-shot predictions, and after fewer than 1000 steps of finetuning with SDS is able to produce unseen scenes of high quality. We show these results in Fig. 8 with 8 objects and 4 styles, where 8 shape-style combinational scenes are masked out during training (opaque scenes in Fig. 8).

## D  OUT-OF-DISTRIBUTION CONVERGENCE

In Fig 5 we show the inference time prompts and the corresponding results. Here we provide the list of prompts used to train the model: "Stained glass chair", "Terracotta chair", "Tiger stripes chair", "Plaid toaster", "Terracotta toaster", "Tiger stripes toaster", "Plaid bowl", "Terracotta bowl", "Tiger stripes bowl", "Stained glass couch", "Plaid couch", "Tiger stripes couch", "Stained glass pot", "Terracotta pot", "Tiger stripes pot", "Stained glass vase", "Plaid vase", "Tiger stripes vase", "Stained glass table", "Plaid table", "Terracotta table".

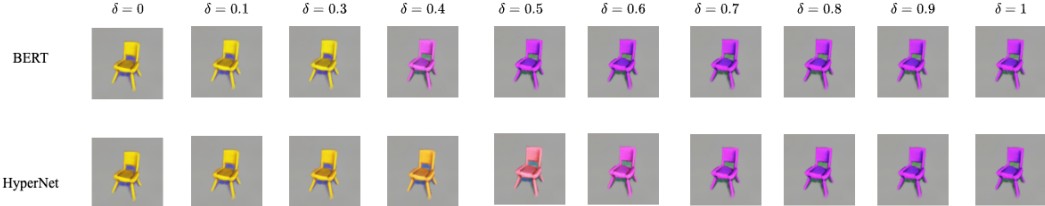

Figure 10: **BERT Token Interpolation.** We show results of interpolating the BERT tokens corresponding to the prompts "yellow chair" and "purple chair". In contrast, interpolation on the level of the hypernetwork ("HyperNet") is smoother than interpolating the BERT tokens.

Since the training prompts dont contain shapes such as "Blender", "Knife", "Skateboard", "Shoes", "Strawberry", "Lamppost", "Teapot" and attributes such as "Gold", "Glacier", "Yarn", "Victorian", we term the prompts used in Fig 5 as out-of-distribution prompts–as the model does not see these shapes and attributes during training.

## E   USER STUDY

In order to get a quantitative evaluation of our generation quality for the out-of-distribution prompts we conduct a human study where we ask participants to rank the render that best adheres to the given prompt in descending order (best render is ranked 1). We compare our method's generation with 33 different DreamFusion models. 1 is trained from scratch and the other 32 are finetuned from checkpoints corresponding to the prompts in section  D. Of these 33 models we pick the best model for each of the out-of-distribution prompts, so the computational budget to find the best baseline for a given prompt is almost 33x that our of method.  Note each of these models, including ours, are trained for the same number of steps. We report average user-reported rank for our method and the average best baseline chosen *for each prompt* in Tab. 2. We outrank the best DreamFusion baseline consistently across all our out-of-distribution prompts.

| Model | Golden Blender | Yarn Shoes | Yarn Skateboard | Plaid Skateboard | Plaid Lamppost | Strawberry with tiger stripes |
|---|---|---|---|---|---|---|
| Our Method (↓) | 1.30 | 1.07 | 1.32 | 1.29 | 1.36 | 1.12 |
| Best DreamFusion Baseline (↓) | 2.50 | 2.40 | 2.33 | 1.75 | 2.0 | 2.25 |

Table 2: **Average User-Reported Ranks (N=450):** We report the average rank submitted by all users for our method, and compute the average rank for all 33 of the baselines. We report the average rank of the best performing baseline for each prompt. Our method is consistently preferred over the best baseline, despite the best baseline consuming 33x more computational resources than our method to find.

## F   BERT TOKEN INTERPOLATION

Another option for interpolation is to interpolate the input BERT embeddings fed in the our Dynamic HyperNet. We show results in Fig. 10 where we interpolate across two chair colors in a Dynamic HyperNet trained on only chair colors. The interpolation is highly non-smooth, with a single intermediate color shown at $\delta = 0.4$ and discontinuities on either end at $\delta - 0.3$ and $\delta = 0.5$. On the other hand, our HyperNet token interpolation shown in Figure 10 demonstrates a smooth and gradual transition of colors across the interpolation range. This demonstrates that our HyperNet learns a smoother latent space of NeRFs than the original BERT tokens correspond to.

## G   HYPERFIELDS WITH PROFLIC DREAMER TEACHERS

We demonstrate that HyperFields is capable of learning several high quality scenes with complex prompts. Following the training pipeline in Fig. 2, in stage 1 we first train a separate Prolific Dreamer model for each of prompt we want the HyperFields model to learn (Wang et al., 2023).  After training the separate Prolific Dreamer model, in stage 2 we use them as teacher model and train our

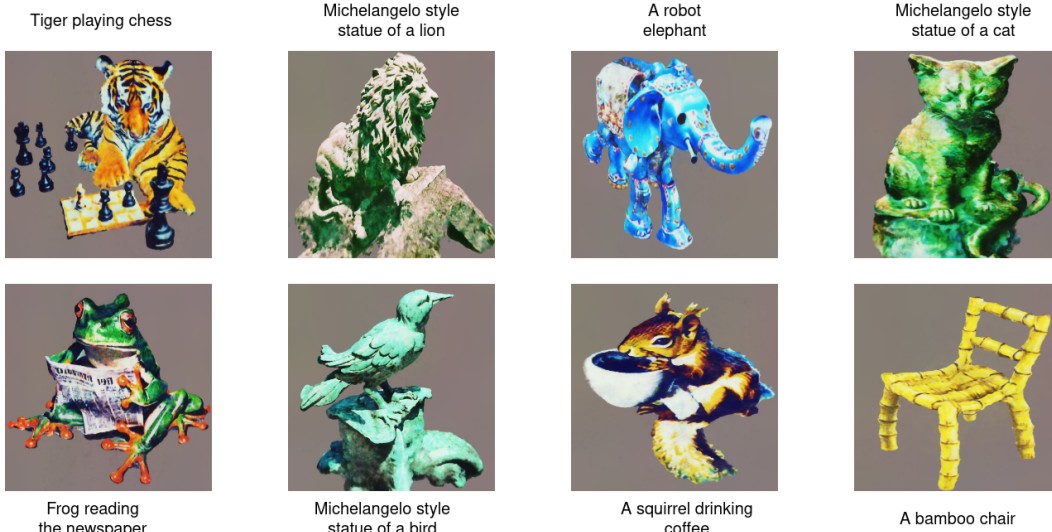

Figure 11: **Distilling Prolific Dreamer into HyperFields:** Demonstrates that HyperFields can be made to learn high quality complex scenes simply by using better teacher model like Prolific Dreamer.

HyperFields model by distilling the teacher Prolific Dreamer into HyperFields. Consequently, we see that HyperFields is able to learn several high quality complex scenes. In Fig. 11, we show a handful of the several scenes learned by a single HyperFields model.

In addition to the scenes in Fig. 11, we show 30 more scenes learned by a single HyperFields model in Fig. 12. We train a single HyperFields model to learn in total 50+ scenes (we show 38 of those scenes Fig. 11, Fig. 12)

## H ADDITIONAL ABLATION ON DYNAMIC HYPERNETWORK

We train both the model used in figure 13 simultaneously on prompts 'Wooden Table' and 'Origami Chair', we see that our model easily learns to generate both these scenes, while the model without the dynamic hypernetwork suffers from lack flexibility and consequently make the geometries of the 'Wooden table' and 'Origami Chair' similar.

## I ALGORITHM FOR TRAINING HYPERFIELDS

---

### Algorithm 1: Training HyperFields with NeRF Distillation

---

**Require:** $\mathcal{T} = \{\mathcal{T}_1, \mathcal{T}_2, \cdots \mathcal{T}_N\}$        ▷ Set of text prompts
**Require:** $\mathcal{C}$        ▷ Set of Camera view points
**Require:** $\theta_1, \theta_2, \cdots \theta_N$        ▷ pre-trained NeRFs
**Require:** $\phi_{HF}$        ▷ Randomly initialized HyperFields
**Require:** $\mathcal{R}$        ▷ Differentiable renderer function
  1: **for** each step **do**
  2:      $\mathcal{T}_l, \mathcal{T}_m, \mathcal{T}_n \sim \mathcal{T}$        ▷ Sample text prompts from $\mathcal{T}$
  3:      **for** $\mathcal{T}_i \in \{\mathcal{T}_l, \mathcal{T}_m, \mathcal{T}_n\}$ **do**
  4:          $\mathcal{C}_i \sim \mathcal{C}$
  5:          $\mathcal{I}_i = \mathcal{R}(\theta_i(\mathcal{C}_i))$        ▷ $i^{th}$ nerf renders image for given camera $\mathcal{C}_i$
  6:          $\mathcal{I}_i{'} = \mathcal{R}(\phi_{HF}(\mathcal{T}_i, \mathcal{C}_i))$        ▷ Condition $\phi_{HF}$ on $i^{th}$ prompt
  7:          $\mathcal{L}_i = (\mathcal{I}_i - \mathcal{I}_i{'})^2$
  8:      **end for**
  9:      $\mathcal{L}_d = \sum\limits_{i \in \{l,m,n\}} \mathcal{L}_i$
10: **end for**

---

## J ADDITIONAL OUT-OF-DISTRIBUTION RESULTS

To add more variability to our OOD (out-of-distribution) results from Fig. 5 we add animal prompts. We consider 'wooden squirrel' and 'squirrel playing flute', for both these prompts the geometry

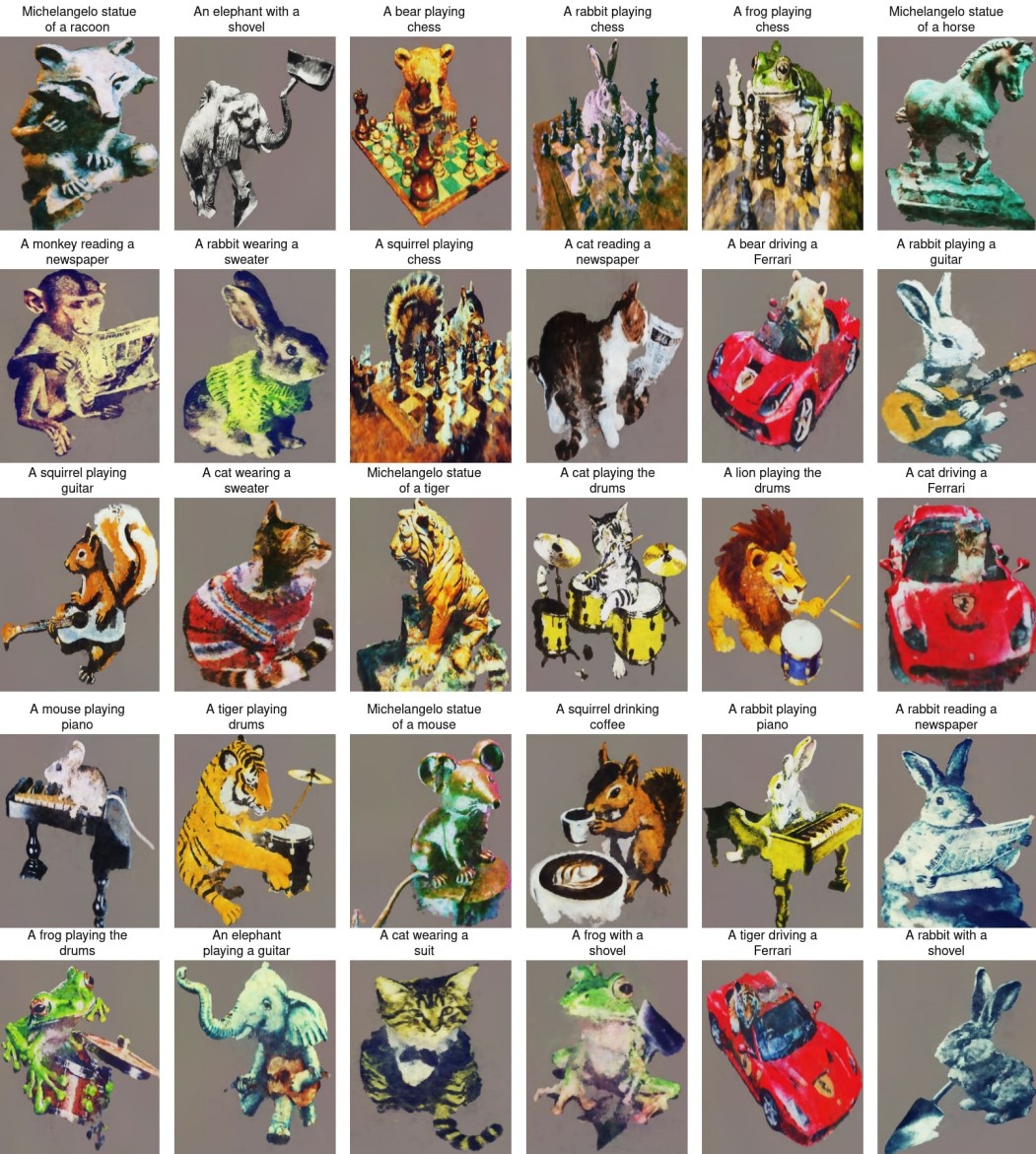

Figure 12: **Additional Prolific Dreamer scenes distilled into HyperFields:** depicting various complex poses of animals, thereby underscoring the ability of a single HyperFields model to learn multiple complex scenes.

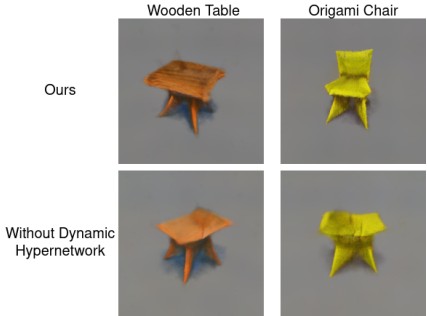

Figure 13: **Ablation over dynamic hypernetwork:** We see that our dynamic hypernetwork can generate origami chair and wooden table, while when trained without our dynamic hypernetwork the model suffers from mode collapse and makes the origami chair have a geometry similar to the wooden table.

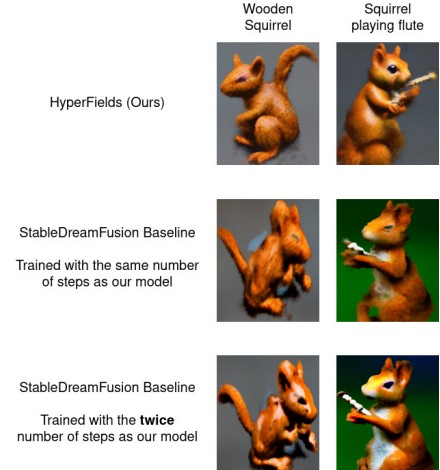

Figure 14: **Ablation over dynamic hypernetwork:** We see that our dynamic hypernetwork can generate origami chair and wooden table, while when trained without our dynamic hypernetwork the model suffers from mode collapse and makes the origami chair have a geometry similar to the wooden table.

generated by HyperFields seems to be better than the StableDreamFusion outputs in our opinion. The corresponding renders are in Fig 14. These prompts are OOD because the model has not seen the attribute wooden and playing flute in its training set. We optimize our HyperFields model using SDS loss.

