# OpenReview forum: "HyperFields: Towards Zero-Shot Generation of NeRFs from Text"
_ICLR.cc/2024/Conference — Submitted to ICLR 2024_

### Official Review · Reviewer_VyWD · 2023-10-30

**Soundness:** 3 good
**Presentation:** 2 fair
**Contribution:** 3 good
**Rating:** 5
**Confidence:** 5

**Summary:**

The paper presents "HyperFields", an approach designed to achieve zero-shot generation of Neural Radiance Fields (NeRFs) from textual prompts. By utilizing a dynamic hypernetwork, HyperFields is able to establish a mapping from text token embeddings, specifically derived from BERT, to the domain of NeRF parameters. The paper introduces NeRF distillation training, where individual NeRF-encoded scenes are distilled into a singular dynamic hypernetwork. The main goal of HyperFields is to efficiently generate scenes that the model has seen during its training (in-distribution), and to quickly fine-tune itself for unseen (out-of-distribution) prompts, if necessary.

**Strengths:**

(+) The paper addresses an important gap in zero-shot generation using NeRFs.

(+) The presentation of the method is clear, with a well-structured explanation of the hypernetwork and NeRF distillation.

(+) The paper introduces a fresh perspective on text-to-NeRF synthesis, leveraging hypernetworks, which is a less explored territory in this context.

**Weaknesses:**

(-) **Limited experimental results**: The experimental results presented in the paper, particularly in Figure 4 and Figure 5, are quite limited. The in-distribution generalization showcased in Figure 4 uses only 9 basic shapes, and the generalization is restricted to simple uniform color translations. Figure 5, too, is restricted to basic geometry and appearance.

(-) **Low quality of results**: The results presented, especially the simple geometric shapes and uniform color distributions, seem to be of low standard. There is a noticeable disparity in quality when compared to state-of-the-art techniques. Moreover, the ablation study in Figure 6 only provides a single example, which weakens the overall argument.

(-) **Lack of comparative discussion**: It's important to note that recent studies, such as HyperDiffusion (ICCV 2023) and Shap-E (arXiv), have also explored the use of hyperparameters in 3D object generation, resulting in promising results. However, there is a lack of comparative discussion with these methods, which is crucial in positioning HyperFields in the current research landscape.

**Questions:**

- In Section 4.2 (referencing Figure 5), the paper mentions, "...with no optimization, when provided with the out-of-distribution prompt. The model chooses the semantic nearest neighbor from its training data as the initial guess for out-of-distribution prompts..."; However, it is not clear how the model is capable of retrieving the nearest-neighbor. Could the authors provide more information on the intrinsic capabilities of the model that enable this nearest-neighbor retrieval?
- How does HyperFields handle highly creative textual prompts?
- How does the NeRF distillation process handle the potential loss of scene details?
- It is important to have a discussion regarding the limitations of the proposed approach.

---

> ### Author Response · Authors · 2023-11-22
> **Response to R VyWD**
>
> > Limited experimental results/low quality of results
>
> Please see our results on complex scene generation in the general comment above. The quality of our results are a direct consequence of the quality of the teacher NeRFs we distill.
>
> We provide an additional ablation demonstrating the utility of the dynamic hypernetwork, please see this [figure](https://ibb.co/KNGThj7). In this experiment we train prompts 'wooden table' and 'origami chair' together. We see that our dynamic hypernetwork can generate origami chair and wooden table, while when trained without our dynamic hypernetwork the model suffers from lack of flexibility and makes the origami chair have a geometry similar to the wooden table. This ablation is discussed in Appendix H.
>
> > Lack of comparative discussion to HyperDiffusion and Shape-E
>
> We have updated the related works section of the paper to cite and discuss these papers. We note that HyperDiffusion is an unconditional generative model which diffuses over NeRF weights (cannot do text based scene generation), whereas our method predicts NeRF weights dynamically conditioned on the 1) text prompt, 2) the sampled 3D coordinates, and 3) the previous NeRF activations. Similarly, Shap-E diffuses over latent codes which are then mapped to weights of a NeRF MLP, and requires teacher point clouds to train. Both of these methods have the same limitations of slow inference due to denoising sampling.
>
> > Out-of-distribution nearest neighbor retrieval
>
> BERT encoded representation of the text prompt is given as input to the hypernetwork. So when an out-of-distribution prompt  'blender’ is provided as input to the hypernetwork, the representation of 'blender’ is similar to the representation of 'toaster’--which is a shape the hypernetwork is trained on. Hence, at step zero the pre-trained hypernetwork generates weights of NeRF that render a 'toaster’.
>
> > How does HyperFields handle highly creative textual prompts
>
> Please see our results on complex scene generation in the general comment above, where we demonstrate our model’s capability to generate creative textual prompts.
>
> > How does the NeRF distillation process handle potential loss of scene details?
>
> Our distillation approach provides strong supervision using the teacher NeRFs, and thus our model is able to converge to very low loss values. This translates to virtually no loss in visual quality of the training scenes predicted by the hypernetwork. See relevant figure [linked here](https://ibb.co/4mrrQBQ).

---

> > ### Comment · Reviewer_VyWD · 2023-11-23
> >
> > Thanks for addressing the initial concerns in the rebuttal. The inclusion of additional results and ablations is appreciated and enhances the comprehensiveness of the work. However, upon reviewing these supplementary materials, I still find that the quality of the results appears rough.
> >
> > Furthermore, In my initial review, I did not mention ATT3D due to its very recent introduction at ICCV. However, I align with the opinions of Reviewers YDm2 and 6JFZ regarding the importance of comparing the HyperFields work with ATT3D, especially in terms of methodological novelty and result quality. After reviewing the rebuttal, my concerns in these areas persist.

---

### Official Review · Reviewer_6JFZ · 2023-11-01

**Soundness:** 3 good
**Presentation:** 3 good
**Contribution:** 2 fair
**Rating:** 5
**Confidence:** 4

**Summary:**

This paper propose a fast way to achieve text-to-3D generation. The key idea is to design a hyper-network that predicts weights that process the spatial latent code of NeRF. The technical contribution is that the hyper-network not only looks at the text input, but also the activations of the MLP to be predicted. Experiment results are shown with only combination of color and shape instead of a full prompt (possibly due to limited computational resources). Qualitative results with very limited examples show that the technique is able to improve text-to-3D speed and achieve certain generalizability toward new concepts.

**Strengths:**

- The hyper-network is fast since it’s free of optimization. Such feedforward approach can have computational advantages as the training cost can potentially be amortized
- The experiment shows that the model can compose concepts in certain fashion. This helps illustrate the benefit that such model can amortize training compute to be used for many inference uses.

**Weaknesses:**

- Current methods is trained from scratch, which might be computationally expensive.
- The key architecture of this paper is a hyper-network that predicts the weights for the MLP.
- A main concern regarding the result is very limited. Most of the results are shown in simple objects and compositions.
- An other small concern is regarding the need to create a small dataset of NeRF scenes. Each NeRF can takes minutes, and this prevents it to scale to larger datasets.
- If the model weight depends on where we sample the activations, then the generated weights can have high variance. I’m a bit concerned that this means different ways to sample the points can lead to different weights, and thus leading to different performance.

**Questions:**

- “SDS loss exhibits high variances” - is there any experimental/reference evidence that support it?
- Quality: Figure 4, why is the image looks washed out? Is it because of artifacts from normalization?
- I wonder if the main difference between ATT3D and this paper is whether the hyper-network takes activations?
- Why do we choose three captions during training? Why can’t we spread away these captions throughout different batches?
- Maybe I’ve missed it, but how do we choose different a_i’s?

---

> ### Author Response · Authors · 2023-11-22
> **Response to R 6jFZ**
>
> > Main concern regarding the result is very limited
>
> Please see our results on complex/high quality scenes in the general comment above.
>
> > Scaling NeRFs to larger datasets
>
> We discuss in Sec 4.3 in the paper that our model is able to amortize the overhead cost of both training the NeRF dataset and distilling them into the HyperFields model. Furthermore, from just a handful of NeRFs we are able to bootstrap our model to converge faster on novel unseen prompts with minimal fine-tuning (Fig. 5), thus only aiding in the effort towards building a larger dataset.
>
> > Variance of generated weights due to sampling
>
> Please see [figure linked here](https://ibb.co/GcLMmZW) for our experiment on the effect of the variance of the  generated weights  due to sampling. We query HyperFields on “a red chair” and shuffled the query rays into random mini-batches for the same viewpoint, and show the render outputs above. Note that the renders are visually identical, **even though the weights predicted by the hypernetwork are different as a result of the different mini-batch compositions**. This is because we train the HyperFields model by distilling teacher NeRFs which naturally imposes 3D view consistency constraints. Thus, our model’s high expressive potential through adaptive NeRF weights does not hinder the view-consistency of scenes due to our distillation training.
>
> > SDS loss exhibits high variance
>
> The stochasticity of the denoised output is a fundamental feature of denoising diffusion models, as the outputs of these models are dictated by randomly sampled noise over a sequence of timesteps. For diffusion models, this is a key feature which enables the substantial variation in images generated by these models. Thus, the SDS loss, which is a function of the diffusion denoiser, will similarly be stochastic and exhibit substantial variance. As an example, please see the output of Stable Diffusion after denoising **for the same timestep (980) conditioned on the same input image and text prompt (left) for two different noise samples** [figure linked here](https://ibb.co/yPdzzv0). Note that for any given denoising timestep, the predicted denoised image will vary significantly.
>
> > Figure 4 images look washed out
>
>  We purposely fade some of the renders to indicate that those are the training scenes. Please see a revised version of the figure [linked here](https://postimg.cc/tYHrxNyM) where the feedforward inference scenes are outlined in black dashed lines, and the training scenes have no outline.
>
> > Difference between ATT3D and HyperFields
>
> Please see our general comment at the top.
>
> > Why do we choose three captions during training?
>
> Can the reviewer please clarify this question? We are not sure which implementation detail this question is referring to.
>
> > How do we choose different a_i’s?
>
>  The a_i’s are the activations of layer i of the NeRF MLP for a given sampled 3D coordinate. These activations are fed into the hypernetwork along with the conditioning text embedding to predict the weights for the NeRF MLP layer i+1. Please refer to Fig. 3 [linked here](https://postimg.cc/sBRSnw6k) for a visualization of this process.

---

### Official Review · Reviewer_ckby · 2023-11-01

**Soundness:** 4 excellent
**Presentation:** 3 good
**Contribution:** 4 excellent
**Rating:** 8
**Confidence:** 4

**Summary:**

This paper proposes a novel generalizable text-to-3D framework to generate a 3D representation with neural radiance field (NeRF) given text inputs. The text inputs are processed by a pretrained BERT network to get powerful embeddings to condition the NeRF generation process. The dynamic hypernetwork is the key innovation of the design to make it succeed to generalize across different 3D shapes over various text conditions. Then a distillation loss is employed to train the NeRF image rendering process given 3D coordinates and synthesized 2D images from DreamFusion. This work will facilitate significant progress of 3D AIGC and inspire future explorations on generalizable 3D generation.

**Strengths:**

(1) The motivation of this work is clear and the technical impact of this work are significant. Limited by the intrinsic mapping relationship between 3D coordinate and color field of NeRF, current work struggles to achieve generalizable 3D shapes with various conditional input with a unified framework. This work proposes a hypernetwork architecture, which is justified as the key innovation to learn a generalized text-to-3D mapping across different inputs.
(2) Authors conduct extensive experiments on both in-distribution and out-of-distribution samples during testing, and results look consistently appealing.
(3) Benefit from the use of InstantNGP, the generation is much faster than the baseline DreamFusion, which is crucial for some real-time applications such as real-time rendering and editing.
(4) Authors have committed that they will release all the code and pretrained models, which will facilitate better reproduction and follow-up for the community.

**Weaknesses:**

(1) There seems missing quantitative comparison between the proposed method and baseline (DreamFusion) on CLIP retrieval scores or user study, which may make the work further stronger and more convincing.
(2) Another ablation study to conduct is to verify the effectiveness of training across multiple shapes then fine-tuning on a single shape, and compare it with a baseline that train the model on this single shape from scratch. This will demonstrate the advantage of training across a wider range of samples to learn a stronger representation between text condition and the 3D representation, by incorporating more samples during training.

**Questions:**

I may consider further improve my rating if my concerns (listed in the weaknessed part) are well addressed.

---

> ### Author Response · Authors · 2023-11-22
> **Response to R ckby**
>
> > ... missing quantitative comparison for baseline on CLIP retrieval scores …
>
> Thank you for the suggestion. In section E of the supplementary, we have user studies comparing the quality of the out-of-distribution renders of HyperFields with that of the Stable-DreamFusion baseline. Our method is consistently preferred over the baseline. Please see the table copied below.
>
> | Model | Gold Blender | Yarn Shoes | Yarn Skateboard | Plaid Skateboard | Plaid Lamppost | Strawberry with tiger stripes |
> | ----------- | ----------- | ----------- | ----------- | ----------- | ----------- | ----------- |
> | Our Method (↓)  | 1.30 | 1.07 | 1.32 | 1.29| 1.36 | 1.12 |
> | Best DreamFusion Baseline | 2.50 | 2.40 | 2.33 | 1.75 | 2.0 | 2.25 |
>
> Please see the results below for the results of CLIP retrieval comparing HyperFields with StableDreamFusion baseline. We compute CLIP retrieval scores for out-of-distribution scenes in [Fig. 5](https://raw.githubusercontent.com/threedle/hyperfields/rebuttal/ood_compare_RLfix.jpg) for both Hyperfields and the Stable-DreamFusion (S) baseline, which is the NeRF model trained from scratch.  We significantly outperform the baseline when the baseline and our model are  trained for the same number of steps. We compare favorably when the baselines are trained 5x longer than our model.
>
> The support set of scenes is a cross product of the following geometry and attribute set.
> - Geometry: {‘chair’, ‘pot’,’toaster’, ‘bench’, ‘stool’, ‘bowl’, ‘vase’, ‘couch’,’table’}
> - Attributes: {‘bamboo’, ‘brick’, ‘yarn’, ‘glacier’, ‘gold’, ‘origami’, ‘plaid’, ‘rainbow’, ‘stained glass’, ‘terracotta’, ‘wooden’}
>
> The support set has 108 scenes in total. We perform CLIP retrieval on this support set.
>
> | Model | Top-1 | Top-10 |
> | ----------- | ----------- | ----------- |
> | Stable DreamFusion (Same compute budget at our model)  | 0.12 | 0.25 |
> | Stable DreamFusion (Trained for 5x longer than our model)  | 0.62 | 0.75 |
> | HyperFields (ours)  | 0.75 | 0.87 |
>
> For additional quantitative evaluation please refer to Table 1 in the paper. We compare the CLIP retrieval scores of the scenes generated by HyperFields for prompts it has seen during training and prompts it generated zero-shot with no optimization. Our results suggest that the quality of the zero-shot rendered novel scenes is similar to that of the quality of the scenes seen during training. For additional discussion please refer to section 4.1 in paper.
>
> > ... verify effectiveness of training across multiple shapes then fine-tuning … and compare with baseline that trains from scratch …
>
> The effectiveness of training across multiple scenes then fine-tuning is demonstrated in [Fig. 5 in the paper](https://ibb.co/QY2TkHS).  We demonstrate that by training across multiple scenes, our model learns a meaningful map between text embeddings and NeRF, such that when fine-tuning to an out of distribution scene (prompt for which the attribute/shape tokens are not seen during training) our model converges significantly faster (~5x) than a model trained from scratch. Specifically, the comparison to the baseline model trained from scratch is given by rows 2 and 4 in [Fig. 5](https://ibb.co/QY2TkHS).

---

### Official Review · Reviewer_YDm2 · 2023-11-02

**Soundness:** 3 good
**Presentation:** 3 good
**Contribution:** 2 fair
**Rating:** 5
**Confidence:** 4

**Summary:**

This work introduces a framework for achieving zero-shot NeRF generation from texts. This is accomplished through the training of a dynamic hypernetwork using hundreds of pre-trained NeRFs to acquire the mapping from text token embeddings to NeRF weights. Extensive experiments demonstrate the capability of the proposed method to predict in-distribution scenes in a zero-shot manner and out-distribution scenes with a few steps of fine-tuning.

**Strengths:**

1. This paper is clearly written and easy to follow.

2. The proposed pipeline intuitively makes sense and it is interesting to see the disentanglement of different attributes learned by the proposed hypernetwork.

**Weaknesses:**

1. My primary concern is the technical contributions of this work in comparison to the referenced ATT3D study. Specifically, while this paper clarifies the connections with ATT3D, it remains unclear what novel techniques or insights are newly introduced by this work. A more compelling justification is highly desirable.

2. Furthermore, there is a lack of benchmarking against ATT3D and the reported results indicate that ATT3D may achieve much better visualization effects. This discrepancy arises because the proposed method appears to only disentangle objects with simple attributes like colors, while ATT3D's reported visualizations can manipulate higher-level attributes and behaviors, such as "chimpanzee holding a cup." The authors are highly expected to conduct a benchmarking comparison with ATT3D using the same text prompt.

3. Although intriguing, it remains unclear why the hypernetwork can successfully learn the disentanglement of various attributes. This may be attributed to the limited scope of text prompts and attributes during training. The authors are expected to provide more insights on this matter.

**Questions:**

My questions are included in the weakness section. I am willing to adjust my scores if my concerns are properly addressed.

---

> ### Author Response · Authors · 2023-11-22
> **Response to R YDm2**
>
> > ATT3D
>
> Please see our discussion of ATT3D in the general comment.
>
> > ... unclear why the hypernetwork can successfully learn the disentanglement of various attributes ....
>
> The disentanglement is implicitly enforced through the training, as the model sees instances of the same text token (e.g. “plaid”) in association with different shapes (chair, toaster, table, etc.), and thus must learn the common features which define a “plaid” style across these different scenes. The same reasoning applies to the object/shape tokens shared across different scenes. If the model attempts to correlate any attribute token with shape token (e.g. “plaid” with “chair”), then it will be penalized when it encounters a prompt like “origami chair” and predicts plaid attributes for the chair, thus forcing it to disentangle the different attributes.

---

### Author Response · Authors · 2023-11-22
**General Comments**

In general the reviewers agreed that the paper’s method is clear, the problem of feedforward NeRF generation and accelerated convergence to out-of-distribution scenes are well motivated, and that the application of hypernetworks is interesting. The primary concerns raised by the reviewers are: (1) the limited quality/complexity of the scenes presented in the experiments, and (2) the relation of the work with respect to ATT3D. We address these concerns in an updated version and summarize these updates below.

# Scene quality/complexity

Please refer to Fig. 11  in the supplemental (or see the figure link [here](https://ibb.co/YbYnF7C)) for an example of single  HyperFields model learning over high quality, complex scenes, by simply using a better teacher model such as ProlificDreamer. Our method’s generation quality is determined by the quality of the teacher NeRFs (see Fig. 2 in paper). The teacher NeRFs we use for the paper were generated with Stable-DreamFusion, which was the best publicly available model at the time of research. However, our method is general and capable of using any model as teacher NeRFs, and we can distill multiple scenes into a single HyperFields network all while maintaining the level of quality of the underlying model teacher NeRFs.

In addition to the scenes in Fig. 11 in the supplemental (or see the figure link [here](https://ibb.co/YbYnF7C)), we show 30 more scenes learned by a **single HyperFields** model in Fig. 12 in the supplement (or see the figure link [here](https://postimg.cc/QVVwy3x1)). In the limited time available to us we trained a **single HyperFields** model to learn in total 50+ scenes (we show 38 of those scenes Fig. 11, Fig. 12) depicting various complex poses of animals, thereby underscoring the ability of a single HyperFields model to learn multiple complex scenes.

### OOD Results
To add more variability to our  OOD (out-of-distribution) results from Fig. 5 we add OOD animal prompts. We consider ‘wooden squirrel’ and ‘squirrel playing flute’, for both these prompts the geometry generated by HyperFields seems to be better than the baseline StableDreamFusion. The corresponding renders are in Fig 13 Appendix J  and in this [link](https://ibb.co/fd64kVN). These prompts are OOD because the model has not seen the attribute wooden and playing flute in its training set. We optimize our HyperFields model using SDS loss.

# ATT3D

A direct comparison against ATT3D is not possible because neither the model nor the code has been made public. ATT3D’s visual quality and scene complexity is a consequence of  the underlying text to 3D model, Magic3D, which uses proprietary diffusion models eDiff-I and Latent Diffusion Model (LDM) as guidance. These models are trained on far more resources (data & compute) than the publicly available StableDiffusion model we rely on for guidance.

We again emphasize that **with a stronger diffusion model** HyperFields is capable of packing and generating scenes of equivalent quality and complexity, which we demonstrate in the above [linked figure](https://ibb.co/YbYnF7C) and Section G of the supplemental.



## Novel technical contributions

1. We demonstrate the ability to generate out-of-distribution scenes with very few fine-tuning steps (Figure 5 in paper), whereas ATT3D only shows results on in-distribution scenes (convex combinations of training scenes). The ability to generate out-of-distribution scenes with minimal fine-tuning is key as it greatly saves time **on new out-in-the-wild prompts**, not simply prompts restricted to recombinations of the training set.

2. Furthermore, our proposed dynamic hypernetwork architecture is novel, as the predicted NeRF weights are generated based on the activations from the previous layers, leading to better flexibility of our model.  This is key to our ability to pack multiple scenes with drastically varying geometric and color attributes (Ablation Fig 6a in the paper). On the contrary, ATT3D keeps the weights of the NeRF the same, and simply modulates the features of the Instant-NGP grid based on the text prompt. By keeping the MLP weights fixed and only updating the feature grid per prompt, they take advantage of spatial locality of common features across scenes. But it also induces a minor mode collapse, in which common features will be shared among unrelated scenes (e.g. the clothes/style objects in [Fig. 8 from ATT3D](https://research.nvidia.com/labs/toronto-ai/ATT3D/images/generalization_videos/animals_amortized_testing_12.5.mp4) are either entirely black/red, in contrast to the significant visual variation in the [per-prompt optimization](https://research.nvidia.com/labs/toronto-ai/ATT3D/images/generalization_videos/animals_per_prompt_test.mp4)). Our results in (Fig. 7) exhibit greater variation in color palettes, object poses, and geometric detail.

---

### Meta-Review · Area_Chair_B7Tg · 2023-12-10

**Metareview:**

**Summary**


This work achieves zero-shot text-to-NeRF generation with a dynamic hypernetwork. The training is to predict the weights of pre-trained NeRFs given the corresponding textual embeddings. The experiments show the proposed method can generate scenes that the model has seen during its training (in-distribution) and can quickly fine-tune itself for unseen (out-of-distribution) prompts, if necessary.



**Strengths**

1. Both the motivation and writing is clear and easy to follow.
2. Experiments on both in-distribution and out-of-distribution samples validate the proposed hypernetwork approach.
3. As the method is based on hyper-network, there is no need for test-time optimization, which gains a much more speed advantage against DreamFusion.
4. The disentanglement experiments show the compositionality of the learned model.
5. The design of the hypernetwork is novel — the weights are predicted based on text and the activations of previous layers, potentially leading to better flexibility and generalization.


**Weaknesses**

1. The evaluation of the paper seems limited as most of the results are shown in simple objects and compositions. It would be much better to show the scalability on larger, more diverse sets. Also, the results are relatively low quality as concerned by reviewers.
2. The robustness of the proposed weight prediction requires additional quantitative analysis.
3. The claim “SDS loss exhibits high variances” seems not properly explained. SDS loss is known to be mode-seeking and will easily lead to collapsed output.

**Comparison to ATT3D**

The proposed method is very similar to ATT3D, a paper Arxiv online in June that also learns a deterministic mapping of text-to-NeRF generation and uses score-distillation (SDS) as the training objective. Based on the ICLR policy, comparing with arxiv papers after May 28 is not required as they can be treated as concurrent. Also, an exact comparison seems impossible due to the availability of the T2I model.
It is still suggested that some comparisons can be conducted using a fair re-implementation with the same T2I model and text prompts.

**Justification For Why Not Higher Score:**

The biggest concern raised by the reviewers is the low quality of the generated outputs which have clear gaps between Dreamfusion and other existing text-time optimization approaches.  The authors claim that the low quality is due to the usage of T2I models compared with some concurrent works, and show additional results in the Appendix with a stronger "ProfilicDreamer" teacher.
However, these claims are also a bit hard to verify as StableDiffusion has already been widely used in various text-to-3D applications such as StableDreamfusion, and the distilled results also seem not satisfying (only a few scenes, and unclear if the model can generalize to unseen prompts).
To demonstrate the significance of the dynamic part of the proposed method, it may be better to conduct on larger sets of text prompts.

**Justification For Why Not Lower Score:**

N/A

---

### Decision · Program_Chairs · 2024-01-16

Reject